# In Silico Prediction of Metabolic Reaction Catalyzed by Human Aldehyde Oxidase

**DOI:** 10.3390/metabo13030449

**Published:** 2023-03-19

**Authors:** Mengting Huang, Keyun Zhu, Yimeng Wang, Chaofeng Lou, Huimin Sun, Weihua Li, Yun Tang, Guixia Liu

**Affiliations:** Shanghai Frontiers Science Center of Optogenetic Techniques for Cell Metabolism, Shanghai Key Laboratory of New Drug Design, School of Pharmacy, East China University of Science and Technology, Shanghai 200237, China

**Keywords:** sites of metabolism, human aldehyde oxidase, drug metabolism, QSAR, machine learning

## Abstract

Aldehyde oxidase (AOX) plays an important role in drug metabolism. Human AOX (hAOX) is widely distributed in the body, and there are some differences between species. Currently, animal models cannot accurately predict the metabolism of hAOX. Therefore, more and more in silico models have been constructed for the prediction of the hAOX metabolism. These models are based on molecular docking and quantum chemistry theory, which are time-consuming and difficult to automate. Therefore, in this study, we compared traditional machine learning methods, graph convolutional neural network methods, and sequence-based methods with limited data, and proposed a ligand-based model for the metabolism prediction catalyzed by hAOX. Compared with the published models, our model achieved better performance (ACC = 0.91, F1 = 0.77). What’s more, we built a web server to predict the sites of metabolism (SOMs) for hAOX. In summary, this study provides a convenient and automatable model and builds a web server named Meta-hAOX for accelerating the drug design and optimization stage.

## 1. Introduction

In the process of drug development, it is important to predict the activity of multiple metabolic enzymes, not only cytochrome P450 (CYP450) but also non-P450 enzymes such as conjugated enzymes and aldehyde oxidase (AOX, EC 1.2.3.1). Aldehyde oxidase has recently gained increased attention in the drug discovery process and the number of drug candidates that are metabolized by AOX is steadily growing [1].

The difference in AOX subtypes among various species is of great significance in drug metabolism research [2,3]. It was responsible for a series of drug failures in clinical trials that AOX-catalyzed biotransformation caused metabolic inactivation and toxicity, and there were some examples shown in Appendix A. FK3453 is a new type of adenosine A1/2 dual inhibitor used in the treatment of Parkinson’s syndrome [4]. Although the results of pre-clinical animal experiments are exciting (bioavailability of more than 30% in rats and dogs), the FK3453 project was terminated due to extremely low human system exposure in clinical trials. Thus, the metabolism of FK3453 in rats cannot reflect its metabolism in humans [5]. RO1 is a novel p38 MAP kinase inhibitor for the treatment of rheumatoid arthritis. The clinical development of RO1 was terminated because of its unexpectedly rapid clearance in human subjects. Its short half-life and metabolic characteristics in humans are very different from those in rats, dogs, and monkeys in conventional preclinical studies [6]. SGX-523 is an orally bioavailable, ATP-competitive small molecule inhibitor of MET [7]. JNJ-38877605 is a c-Met Tyrosine Kinase Inhibitor used in anticancer treatment [8]. Both SGX-523 and JNJ-38877605 are metabolized by AOX into insoluble metabolites, resulting in renal toxicity. It is worth noting that when a candidate drug is mainly eliminated by AOX metabolism, the mouse and rat strains most commonly used in drug metabolism studies cannot mimic the processes in the human body. In extreme cases, the species difference in the AOX metabolism may even lead to the failure of the entire drug development project. Thus, it is necessary to build a model to predict the metabolism of human aldehyde oxidase (hAOX).

The impact of hAOX on drug development is significant and previous studies have been reported to be made for developing computational tools for drug metabolism prediction of hAOX. Inspired by the catalytic mechanism of the hAOX-mediated metabolism [9,10], several in silico models have been reported to predict where its potential sites of metabolism (SOMs) locate in the structure. Torres et al. applied density functional calculations to predict the regioselectivity of drugs and molecules metabolized by aldehyde oxidase [9]. Montefiori et al. developed fast methods for the prediction of SOMs mediated by aldehyde oxidase: (1) using ChemBioDraw to calculate the chemical shift or (2) calculating ESP charges or chemical shifts with density functional theory [11]. Zhao et al. combined traditional quantum chemical calculations and decision tree models to predict the SOMs mediated by hAOX [12,13]. It is difficult for these above-mentioned methods to automate and it is time-consuming. Importantly, there is currently no effective and convenient tool for metabolism prediction catalyzed by hAOX.

Fingerprint-based quantitative structure–activity relationship (QSAR) methods have been used for metabolism prediction and achieved good performance in our previous work [14,15]. Meanwhile, a graph-convolutional neural network has been used for small-scale reaction prediction [16,17] and achieved good performance. Graphs were applied to describe molecular structures: atoms were represented as nodes, and bonds were represented as edges [18]. The transformer is a neural machine translation method established by Google, which treats the metabolism prediction problem as a sequence translation problem based on the simplified molecular input line entry system (SMILES) representation of molecules, and some chemical reaction prediction studies applied the transformer method on small-scale data [19,20]. Thus, as shown in Figure 1A, we compared three machine learning methods including the fingerprint-based method, graph-based method, and sequence-based method to predict the metabolic reaction for hAOX which is small-scale but important. We clearly mention that our work aims to show the predictive power of three ligand-based machine learning models in the face of a lack of training data. The overview of the research was shown in Figure 1 and more details of Figure 1 were available at materials and methods. Compared to previous work, our ligand-based method could achieve better results; besides, we provided a web server named Meta-hAOX (http://lmmd.ecust.edu.cn/MetahAOX (accessed on 15 December 2022)) for the SOMs prediction for hAOX, which is beneficial for the drug design and optimization stage.

## 2. Materials and Methods

We employed three ligand-based machine-learning methods to build SOM prediction models for hAOX. They are fingerprint-based, graph-based, and sequence-based methods, respectively. The fingerprint-based method was trained by the Scikit-learn python package (https://scikit-learn.org/0.24/ (accessed on 15 December 2022)) [21]. The graph-based method was trained by the DGL-LifeSci python package (https://github.com/awslabs/dgl-lifesci (accessed on 15 December 2022)) [22], running on the GPU version of the PyTorch framework (https://pytorch.org/ (accessed on 15 December 2022)) [23]. The sequence-based method was trained by the OpenNMT framework (https://opennmt.net/ (accessed on 15 December 2022)) based on PyTorch. The web server was constructed by flask framework (https://flask.palletsprojects.com/en/2.2.x/installation/ (accessed on 15 December 2022)) and written by python scripts.

### 2.1. Data Collection

Data were collected from the literature. The training set was collected from the article by Cruciani et al. [24], while the external test set was collected from other references. The information of the data and reference was listed in Appendix A. As shown in Appendix A, the training set included 198 substrates and the test set contained 53 substrates. All chemical structures were converted into the canonical format and standardized, and duplicates were removed by Pipeline Pilot software. The Principal Component Analysis (PCA) and Tanimoto similarity were employed to calculate the spatial distribution of the training set and test set.

### 2.2. Fingerprint-Based Method

We tried to use the traditional machine learning methods combined with atom environment fingerprints to predict the SOMs for hAOX-mediated reactions. The whole workflow of the traditional machine learning model employed in this work was displayed in Figure 1B. Firstly, we labeled the SOM and non-SOM for the molecule via experimental data. Secondly, three machine learning methods were applied to construct models. Finally, a grid search and feature selection were employed to optimize the parameter.

#### 2.2.1. Potential SOMs and Atom Environment Fingerprints

As shown in Table 1, experts summarized two types of potential SOMs metabolized by hAOX and optimized them as the SMILES arbitrary target specification (SMARTS) strings according to reaction mechanism and hAOX metabolites reported in the literature [12]. We employed atom environment fingerprint to describe specific atoms in a molecule [25]. Three atom environment fingerprints were carried out, including Morgan [26], TopologicalTorsion (TT) [27], and AtomPair (AP) [28] in this work. We set the radius at 1–5 for the Morgan fingerprint, and the whole three fingerprints were set size at 256, 512, 1024, and 2048 bits, so we generated altogether 28 different fingerprints descriptors by the open-source RDKit (http://www.rdkit.org/ (accessed on 15 December 2022)). All potential SOMs will be described as atom environment fingerprints. The SOMs reported in the literature were labeled as “1”. As shown in Appendix A, the training set included 637 potential SOM, 322 of which were reported in the literature, and the test set included 168 potential SOM, 56 of which were true SOM.

#### 2.2.2. Feature Selection

In order to avoid those redundant, noisy, or irrelevant features to improve the prediction accuracy of models [29], we used three feature selection methods including variance threshold (VT), select percentile of feature (SPF), and principal component analysis (PCA). Firstly, VT was used to remove all low-variance features. We set the threshold as the default value of 0, and the features that have the same value in all samples would be removed. Then, two different feature selection methods (SPF and PCA) were employed.

SPF selected features according to a percentile of the highest scores. We can see the details at https://scikit-learn.org/stable/modules/classes.html#module-sklearn.feature_selection (accessed on 15 December 2022). PCA is not a typical method for feature selection, but it can be applied for dimensionality reduction. It is applied to decompose a multivariate dataset in a set of successive orthogonal components which explain a maximum amount of the variance. The details of PCA can be seen at https://scikit-learn.org/stable/modules/decomposition.html#pca (accessed on 15 December 2022). The parameters and values of the VT, SPF, and PCA were shown in Appendix A.

#### 2.2.3. Model Building

Three traditional machine learning methods which are Support Vector Machines (SVM) [30], Random Forest (RF) [31], and Gradient Boosted Decision Trees (GBDT) [32] were employed to build the model. A grid search algorithm during 10-fold cross-validation with the purpose of optimizing the parameters on the training set. The hyperparameters of each machine learning method that need to be searched by the grid search algorithm were shown in Appendix A. According to the performance of 10-fold cross-validation, we selected the top-10 baseline models. The VT, SPF, and PCA were employed to select features on the top-10 baseline models. Finally, according to the performance of the 10-fold cross-validation, we selected the best model and evaluated the model on the test set.

### 2.3. Weisfeiler-Lehman Network

The Weisfeiler–Lehman Network (WLN) is a graph-convolutional neural network (GCN) model developed by Coley et al. [33]. We applied the WLN to predict the SOMs of hAOX in this study.

#### 2.3.1. Data Preprocessing

The data of products and reactants were processed as Reaction SMILES. As shown in Table 2, the reaction was represented as Reaction SMILES (reactants>>products), and then the atom-mapping was added to Reaction SMILES by the Reaction Decoder Tool (RDT) [34]. As shown in Table 2, according to atom-mapping reaction SMILES, the model can capture that the SOM is the atom numbered as “2”.

#### 2.3.2. Model Building

As a category of graph convolutional networks, the WLN is adopted to learn the molecular graph isomorphism by embedding the Weisfeiler–Lehman (WL) algorithm [35]. In the GCN model, a metabolic reaction consists of a pair of molecular graphs where Gr and Gm represent the reactant and the metabolite, respectively. According to the metabolic reaction SMILES, the model can capture the information of SOM and non-SOM. A molecular graph G = (V, E) is composed of an atom set (V) and a bond set (E). Relying on the WL algorithm that aggregates flowing information between neighboring atoms, the information about the state features of the atom and structures of adjacent atoms could be updated. In each iteration, the GCN model can obtain information about the local atom environment. The top-k predicted SOMs are ranked according to their scores. The training set was used to train the model, the validation model was used to select the parameters, and the test set was used for evaluation.

The workflow of the WLN to obtain the atom environment features was shown in Figure 1C. The model will extract the information of the atoms and bonds which were described as nodes and edges. Node descriptors including atom type, atom degree, atom explicit valence, atom implicit valence, and aromaticity, and edge descriptors including bond type, whether the bond is conjugated, and whether the bond is in-ring a were considered. The definitions of all the atom and bond features in this work are available in the supporting information (Appendix A).

#### 2.3.3. Tuning Parameters for WLN

It is not trivial to find the best performing set of hyperparameters for a deep neural network. As our previous work mentioned [17], the WLN model has some parameters, such as batch size, learning rate, and iteration layers, that can influence both its training and its architecture. The performance of the model can vary notably depending on the parameters. For the WLN model, we tuned the parameters including learning rate and iteration layers by gird searching. The learning rate was set to 0.0003, 0.001, 0.003, and 0.01, which were the common values of the learning rate. The iteration layers were set as 1, 2, 3, 4, and 5.

### 2.4. Transformer

The sequence-based machine learning method applied a transformer model. Given an input SMILES that represents a molecule, the model predicts the output SMILES which represents the likely metabolites metabolized by hAOX. In this work, to increase the performance of small-scale metabolic prediction, we introduced the transfer learning strategy to transformer models [20,36,37]. The whole workflow of the sequence-based method is shown in Figure 1D. First, since metabolic reaction is a special kind of chemical reaction, pretraining was performed on a large chemical reaction dataset represented by SMILES. The model learned a lot of basic knowledge including the rules of SMILES, which decreased the invalidity of SMILES generated by the model in the process of pretraining. Second, the learned knowledge was successfully transferred to be used on a smaller dataset. Finally, with the chemical skills from the pretraining, the models could output results with increased accuracy after a short and limited training on a small dataset. The training set was used to finetune the model, the validation model was used to select the parameters, and the test set was used for evaluation.

#### 2.4.1. Data Preprocessing

The transformer model, which is based on an encoder–decoder architecture, shows state-of-the-art performance in chemical reaction prediction and retrosynthetic analysis. It treats the metabolic problem as a sequence translation problem. Before training the model, we should process the data. The dataset for pre-training was collected from an open-source patent dataset of chemical reactions (USPTO dataset) [38], which had been used for training models for chemical reaction prediction [39,40,41]. The step of data preprocessing was shown in Table 3. The reactants and products were tokenized by the following regular expression:
Token_pattern = (\ [[^\ ]]+]|Br?|Cl?|N|O|S|P|F|I|b|c|n|o|s|p|\(|\)|\.|=|#||\+|\\\\|\/|:|~|@|\?|>|\*|\$|\%[0-9]{2}|[0-9]).

#### 2.4.2. Model Building

It has been reported that the transformer is a data-driven model [39], so we applied the transfer learning strategy. An example of transfer learning is solving one problem and applying it to a different but related problem. The transformer–baseline model was trained on the small-scale hAOX reactions directly, while the transformer–transfer learning model was pre-trained on a large-scale chemical reaction dataset, and then fine-tuned on a small-scale hAOX-mediated metabolic reaction dataset.

Beam search is an improvement on the greedy algorithm. The algorithm explores all possible characters and keeps the k most likely sequences instead of expanding the predicted sequence by choosing the most likely character. In this study, the model can generate multiple possible metabolites, and the number of metabolites is equal to the beam size. With the increasing beam size, the coverage of the metabolite space will also increase. According to the collected data, we found that most compounds have 1–2 metabolites metabolized by hAOX, so we set the beam size at 5 to obtain the most likely metabolites.

### 2.5. Validation of Model Performance

Six metrics including sensitivity (SE), specificity (SP), accuracy (ACC), F1-score (F1), Matthews correlation coefficient (MCC), and the area under the receiver operating characteristic curve (AUC) were employed to evaluate the performance of the fingerprint-based model in 10-fold cross-validation and test set. These metrics are based on true positive (*TP*), false positive (*FP*), true negative (*TN*), and false negative (*FN*). The first five indicators were calculated using the following equations:(1)$$sensitivity\;\left(SE\right)=\frac{TP}{TP+FN}$$
(2)$$specificity\;\left(sp\right)=\frac{TN}{TN+FP}$$
(3)$$accuracy\;\left(ACC\right)=\frac{TP+TN}{TP+TN+FP+FN}$$
(4)$$F1=\frac{2TP}{2TP+FN+FP}$$
(5)$$MCC=\frac{TP*TN-FP*FN}{\sqrt{\left(TP+FP)(TP+FN)(TN+FP)(TN+FN\right)}}$$

In order to compare with the other two methods, three metrics (Top-1, Top-2, and Top-3 accuracy) were used to estimate the accuracy. A substrate is considered correctly predicted if the atom, being a real SOM, is ranked the first (Top-1), first or second (Top-2), first, second, or third (Top-3). Let ${\hat{f}}_{i,j}$ be the *j*-th predictive result for the *i*-th molecule, *y_i_* be the actual value, and *n_samples_* be the number of substrates, and then, the top-k accuracy can be written as:(6)$$Top-k\;accuracy(y,\hat{f})=\frac{1}{n_{samples}}\sum_{i=0}^{n_{samples}-1}\sum_{j=1}^{k}1({\hat{f}}_{i,j}=y_{i})$$

Validity is applied to assess if the output SMILES string from the transformer model is effective as some SMILES output from the transformer model cannot be represented as compounds. The indicator was calculated as follows.

### 2.6. Comparison with Published Works

We compared our model (Meta-hAOX) with published models including the DT_AOX_ model [13], DTN_AOX_ [12], and three descriptors proposed by Montefiori et al. [11] Because previous work did not provide a web server, therefore, we additionally applied the method for building the best model selected from three ligand-based methods to train on the same dataset collected by Zhao et al.[12], and then compared the same external set collected by Zhao et al. [12] They collected 73 SOMs, less than what we collected. For the fairness and reasonableness of the comparison, we trained and evaluated our model on the same training set and external set they collected.

## 3. Results

Three machine learning methods including the fingerprint-based method (traditional machine learning method), graph-based method (WLN), and sequence-based method (Transformer) were applied to predict hAOX-catalyzed reactions. Comparing the three ligand methods, we intended to select the best model and build a web server (http://lmmd.ecust.edu.cn/MetahAOX (accessed on 18 March 2023)) for the prediction of SOMs mediated by hAOX.

### 3.1. Data Set Analysis

The PCA method was applied to calculate the drug-likeness of the compounds in the training set and test set. The compounds which were FDA-approved drugs were collected from e-Drug3D [42]. As shown in Figure 2A, two-dimensional PCA on FDA-approved drugs and compounds in the training set and test data set on the feature of Morgan fingerprint (size = 1024 bits, radius = 2) was applied to explore the chemical space distribution of the different datasets. The distribution of the test set was roughly within the scope of the chemical space of the training set, which indicated that our model could predict the structure of the test set. Furthermore, we calculated the Tanimoto coefficient based on the Morgan fingerprint (size = 1024 bits, radius = 2) to calculate the similarity of the training set and test set. The histogram of the frequency distribution of Tanimoto similarity for the training set was shown in Figure 2B, and the average Tanimoto similarity of the molecules on the training set was 0.156. The histogram of the frequency distribution of Tanimoto similarity for the test set was shown in Figure 2C, and the average Tanimoto similarity of the molecules on the training set was 0.154. The low Tanimoto coefficients indicated that the molecules used in this study were structurally diverse.

### 3.2. The Performance of Three Machine Learning Methods

#### 3.2.1. Performance of the Fingerprint-Based Method

Eighty-four different models were built using three traditional machine learning methods and three different fingerprints with special lengths. We selected the top 10 baseline models which did not select features on the training dataset; the performance of the top 10 baseline models is shown in Figure 3, and the best model was constructed by the GDBT machine learning method and AP fingerprint with the bits of 1024. Overall, SVM and GDBT methods occupied the top 10 baseline models, while the RF method did not contribute to building the model. The process of feature selection was executed on the top 10 baseline models. Finally, we selected the model with the best performance. The best model was constructed by the SVM machine learning method and AP fingerprint (2048 bits). The feature selection of the best model was SPF, the percentile of which was set as 15. The performance of the best fingerprint-based model on the training set, 10-fold cross-validation, and test set is shown in Appendix A. The AUC values of them were both above 0.85.

#### 3.2.2. Performance of the Graph-Based Method

We tuned the parameters including learning rate and iteration layers by gird searching. As shown in Figure 1C, we trained our model in different iterations to obtain new atom features. As shown in Figure 4, the model would get better performance when the learning rate was set to 0.001 and the iteration layers were set to 3 according to the results of top 1 and top 2 accuracy on the validation set. The final parameters setting of the graph-based model is listed in Appendix A. According to the performance on the validation set, we selected the best model constructed by WLN. As shown in Figure 5, the best graph-based methods on the test set performed also well, and top-1 accuracy was achieved at 0.83.

#### 3.2.3. Performance of the Sequence-Based Method

The parameter applied in the transformer was shown in Appendix A. The performance of transformer–baseline and transformer–transfer learning models on the test dataset is shown in Table 4. The baseline model received an extremely bad result, and the top 1 accuracy on the test set can only reach 0.08, while the validity of the SMILES string can only reach 0.69. There is an example of the predicted results of the transformer–baseline model and transformer–transfer learning model shown in Appendix A. We found that some predicted SMILES strings of the baseline model cannot present molecules. According to transfer learning, the model learned the rules of SMILES, so the validity of the SMILES improved to 0.93. The performance of the transformer also improved a lot via transfer learning. However, the top 1 accuracy indicator was not high, only 0.67.

### 3.3. Comparison of the Methods Each Other and with Others

According to 10-fold cross-validation on the training set, we selected the best model of each machine learning method, respectively, and evaluated them on the test set. As shown in Figure 5, the fingerprint-based model and graph-based model performed better than the sequence-based model. The fingerprint-based model had the best performance (top 1 ACC = 0.85, top 2 ACC = 0.94, top 3 ACC = 0.98). The graph-based model performed slightly worse than the fingerprint-based model (top 1 ACC = 0.83, top 2 ACC = 0.91, top 3 ACC = 0.94). The sequence-based model didn’t perform well (top 1 ACC = 0.57, top 2 ACC = 0.65, top 3 ACC = 0.67). Finally, we chose the best fingerprint-based model to build a web server (http://lmmd.ecust.edu.cn/MetahAOX (accessed on 15 December 2022)). As shown in Figure 6, the webserver consists of four sections: (A) home, (B) about, (C) prediction, (D) contact.

We also compared our model with published models, including the DT_AOX_ model [13], DTN_AOX_ [12], and three descriptors proposed by Montefiori et al. [11]; the performance of the comparison on the external set collected by Zhao et al. [12]. is shown in Table 5. Compared with the published models, our model achieved better performance (SE = 0.77, SP = 0.93, ACC = 0.91, F1 = 0.77). Compared with DTN_AOX_ proposed by Zhao et al., the SE of our model was slightly lower, but other indicators are better than theirs.

## 4. Discussion

### 4.1. Data Analysis

In this study, we compared three advanced ligand-based methods and then selected the best model to build a web server for hAOX-mediated metabolism prediction. It has been reported that AOX contributes to clinical failures by metabolizing molecules with N-containing heterocyclic aromatic rings. The aromatic N is introduced to stabilize the metabolism by CYP450s while consequently increasing the susceptibility to hAOX [43]. Thus, the substrates in this study were N-containing molecules without high coverage of approved drugs as shown in Figure 2A. All the data were collected from the literature, and we collected as many data as possible. Compared with other publications, this study collected extensive and novel data (with 251 substrates and 375 SOMs).

### 4.2. The Analysis of Our Models

As shown in Figure 3, for the fingerprint-based model, we found that different machine learning algorithms and atom environment descriptors were the main driving factors of the different performance. The AP and Morgan fingerprints performed better than the TT fingerprint. The SVM and GDBT methods performed better than the RF method. Fingerprint-based methods applied traditional machine learning models and received a good performance. The graph-based method used a deep neural network model, WLN; however, it has been reported that the WLN model had an excellent performance on small-scale datasets [16,17]. The WLN model indeed achieved satisfactory results in the task of hAOX-mediated SOMs prediction. The transformer has been made for drug metabolism prediction in global prediction in a large-scale dataset [44] and received good results in the large-scale dataset. With the development of the transfer learning strategy, the transformer has been reported to be successfully applied in a small dataset of chemical reaction [20]. The model was pre-trained on the large-scale dataset and then fine-tuned on the small-scale dataset. Unfortunately, the transformer did not perform well in hAOX-mediated metabolism prediction. The same situation was reported in the previous study [16]. The transformer model failed to absorb enough metabolic reaction knowledge from a training dataset of limited size due to its data-driven nature [39]. The performance of the best model on the training set, 10-fold cross-validation, and test set is shown in Appendix A. The AUC values of them were both above 0.85, indicating that our best model had learned the knowledge of hAOX-mediated metabolic reaction.

Some examples of the results predicted by three different models were shown in Appendix A. The metabolite mediated by hAOX of Cinnoline can only be correctly predicted by the transformer. The metabolite of 2-Mercaptopurine can only be correctly predicted by the WLN model. The metabolite of AMG-900 can be correctly predicted by the fingerprint-base model. However, we found that the predicted SOM AMG-900 by the transformer was correct, while the transformer model output the wrong SMILES string. The SMILES error refers to the incorrect prediction of products’ SMILES. There are two types of SMILES errors: (1) invalid SMILES string, (2) valid SMILES while the wrong predicted results. A side effect of formulating reaction prediction as a translation problem is that changes to a single SMILES character can lead to changes in molecular structure or even invalid structure [41]. Some invalid SMILES strings predicted by the transformer are listed in Appendix A. The predicted metabolite of AMG-900 by the transformer is valid, but it is not the ground truth. The five-membered ring replaced the six-membered ring of the true metabolite. SMILES error is a vital reason why the transformer cannot achieve excellent performance.

### 4.3. Comparison of Our Model with Others

We also compared our best model with published models on the same dataset, and our model had better predictive performance (Table 5). Published methods were based on quantum chemistry and molecular docking inspired by the catalytic mechanism, so these models were explainable. Our model demonstrated decent performance and significantly exceeded DT_AOX_ and models constructed by three descriptors proposed by Montefiori et al. However, our model was built by machine learning methods, which is a black box, and the interpretation needs to be improved.

To further validate our model, some compounds which had hAOX-oxidized metabolites detected by liquid chromatography-mass spectrometry (LC-MS) [12], including S7276, S7577, S1129, S2686, S7832, and S7373, were tested in our web server. As shown in Figure 7, the model gave sound prediction results, and the corresponding SOMs of S7276, S7577, S1129, S2686, and S7832 were in accordance with the proposed metabolic region. Molecules were all correctly predicted except S7373 that contains pyridine, which our model considered as a non-substrate. Lepri et al. [45] proposed that pyridine-containing compounds are always stable to hAOX-mediated biotransformation, while the in vitro experimental results demonstrated that some pyridine substituted molecules are also susceptible toward hAOX such as S7373. The prediction of our model for pyridine ring also needs to be improved. Nevertheless, our model still has a good predictive performance in general.

If a molecule is predicted to be a substrate of hAOX by our model, there are 4 strategies for drug design and pharmacokinetic property optimization [46]: (1) blocking hAOX-mediated oxidation; (2) reducing the rate of hAOX-mediated oxidation; (3) using hAOX-mediated oxidative metabolism to design prodrugs; and (4) completely stopping hAOX-mediated oxidative metabolism by optimizing the above ineffectiveness and fully assessing the risk of preclinical studies.

## 5. Conclusions

The lack of data is a vital factor discouraging the application of an end-to-end learning-based method for the task of metabolism prediction. However, some methods such as transfer learning help the learning-based methods to be applied to the small dataset. In this study, we applied three learning-based methods to predict the SOMs of hAOX and compared them to select the best-performing model to construct a web server. Our best method had some improvement compared with the methods based on quantum chemistry and molecular docking. What is more, previous work was difficult to automate, and we have developed a convenient web server that can be used by medicinal chemists, which is beneficial for drug design and development.

## Figures and Tables

**Figure 1 metabolites-13-00449-f001:**
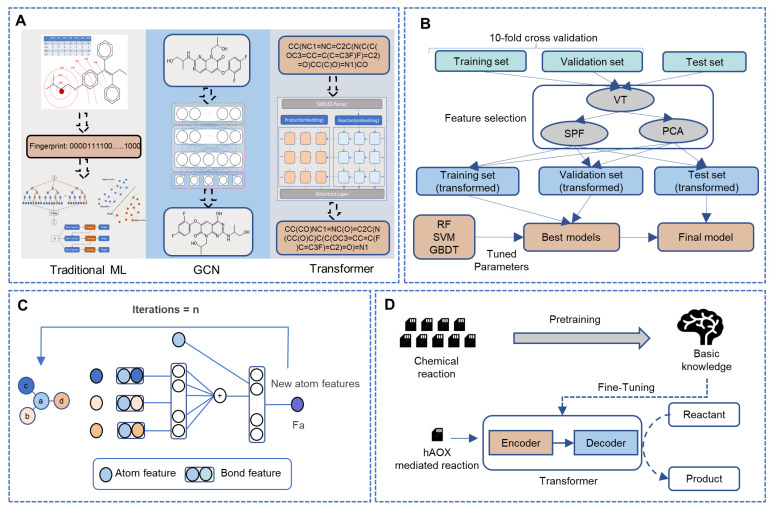
The framework of this study. (**A**) The overview of our work. (**B**) The details of the fingerprint-based method. (**C**) The workflow of the WLN model to obtain the atom environment features. (**D**) The whole workflow of the sequence-based model in this study. (ML: machine learning, GCN: graph convolutional network, VT: variance threshold, SPF: select percentile of feature, PCA: principal component analysis, RF: random forest, SVM: support vector machines, GBDT: gradient boosted decision trees.)

**Figure 2 metabolites-13-00449-f002:**
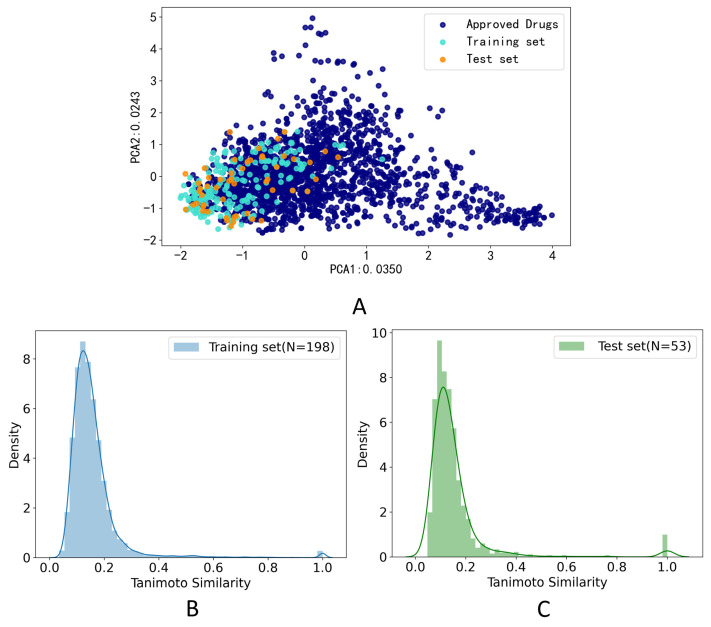
Data distribution. (**A**) Principal component analysis (PCA) on FDA-approved drugs and compounds in the training and test data sets on the feature of Morgan fingerprint (size = 1024 bits, radius = 2); (**B**) Histogram of the frequency distribution of Tanimoto similarity for the training set on the feature of Morgan fingerprint (size = 1024 bits, radius = 2); (**C**) Histogram of the frequency distribution of Tanimoto similarity for the test set on the feature of Morgan fingerprint (size = 1024 bits, radius = 2).

**Figure 3 metabolites-13-00449-f003:**
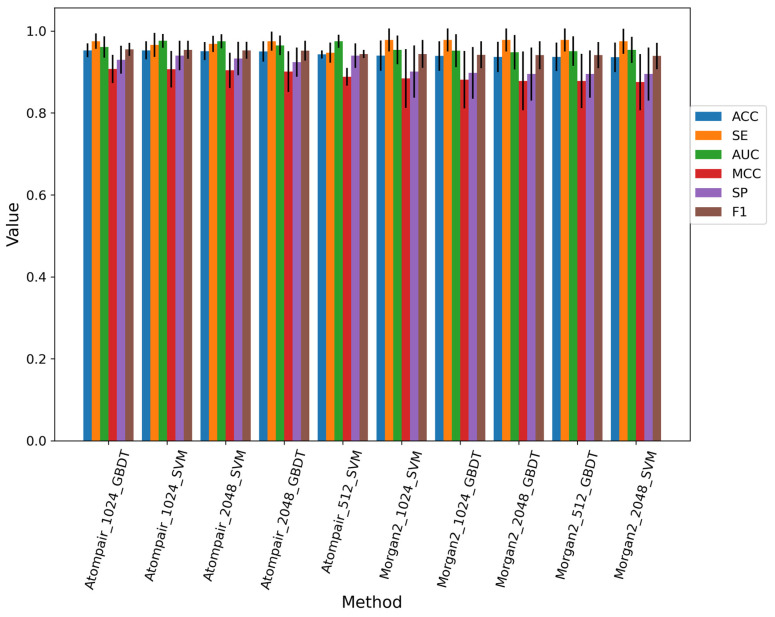
Performance of top 10 baseline models for the fingerprint-based method on the training dataset.

**Figure 4 metabolites-13-00449-f004:**
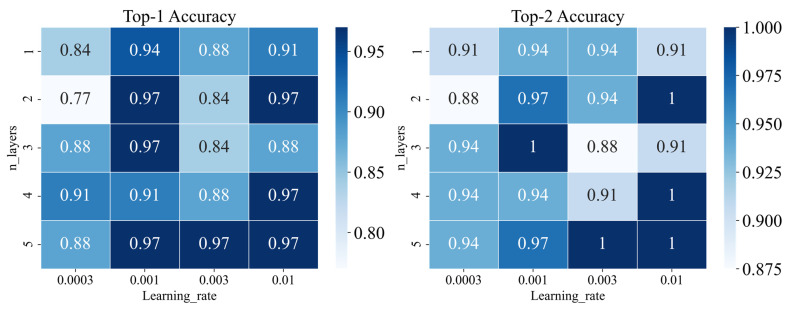
Performance of different parameters (learning rate and n_layers) for the graph-based model on validation set.

**Figure 5 metabolites-13-00449-f005:**
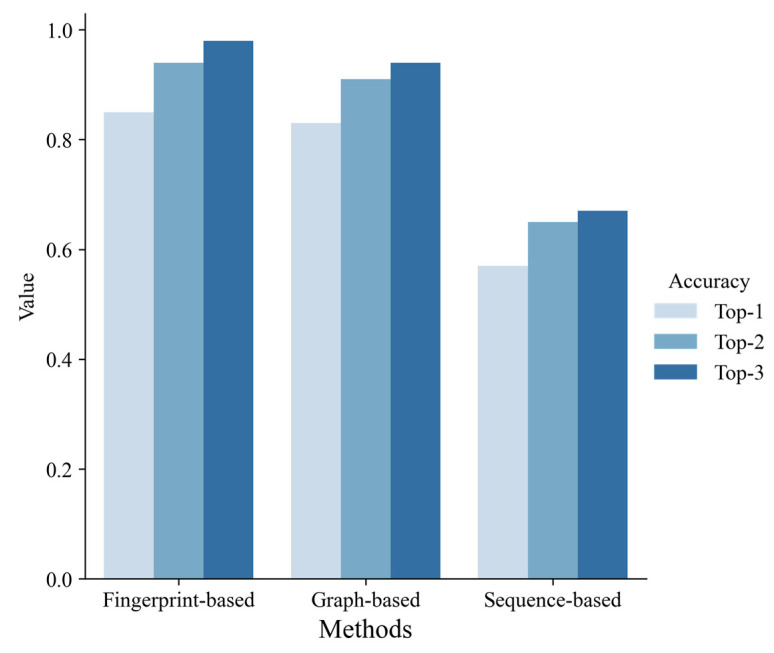
The performance of three different ligand-based methods on the test set.

**Figure 6 metabolites-13-00449-f006:**
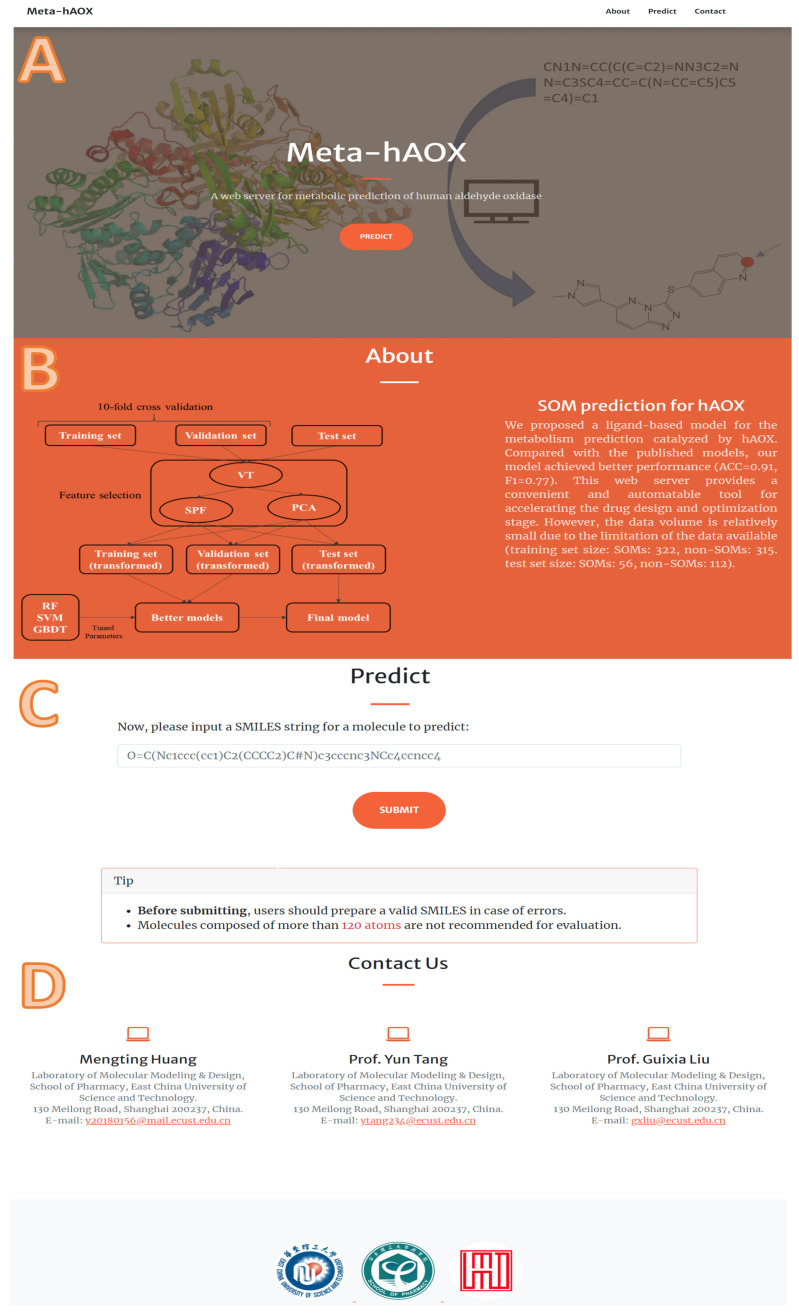
The details of our web server: (**A**) home, (**B**) about, (**C**) prediction, (**D**) contact.

**Figure 7 metabolites-13-00449-f007:**
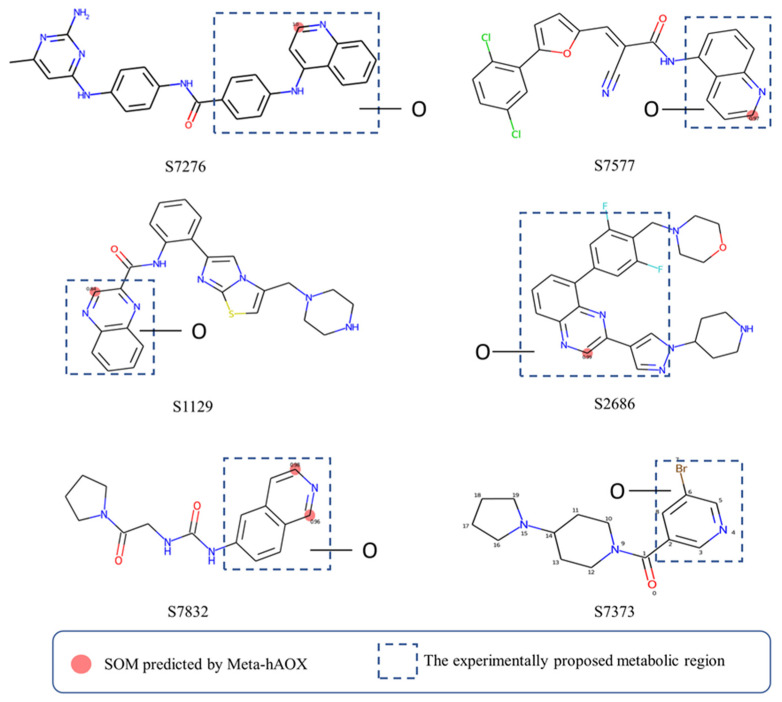
Case study of our model.

**Table 1 metabolites-13-00449-t001:** Potential Sites of Metabolism by hAOX.

Type	SMARTS	Descriptions	Example
A	[$(cR;H]:[nX2R])]	The carbon in the aromatic ring adjacent to the aromatic nitrogen with exactly one hydrogen	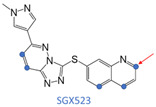
B	[$([#6D2R;H][*][*][#7X2R])]	The carbon in the aromatic ring conjugated addition with γ-position nitrogen with exactly one hydrogen	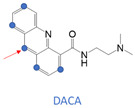

**Table 2 metabolites-13-00449-t002:** Data preprocessing steps of WLN.

Step	Examples: Reactants >> Products
Reaction	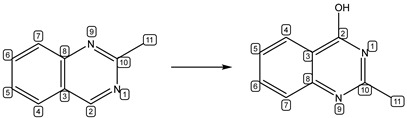
Reaction SMILES	CC1 = NC2 = C(C = CC = C2)C = N1>>CC3 = NC(O) = C4C = CC = CC4 = N3
Atom-mapping Reaction SMILES	[N:1]1 = [CH:2][C:3] = 2[CH:4] = [CH:5][CH:6] = [CH:7][C:8]2[N:9] = [C:10]1[CH3:11] > > [CH3:11][C:10] = 1[N:9] = [C:8]2[CH:7] = [CH:6][CH:5] = [CH:4][C:3]2 = [C:2]([OH:12])[N:1]1

**Table 3 metabolites-13-00449-t003:** Data preprocessing steps of transformer.

Step	Reactant	Product
SMILES	CC(CO)NC1 = NC = C2C(N(CC(O)C)C(C(OC3 = CC = C(F)C = C3F) = C2) = O) = N1	CC(NC1 = NC(O) = C2C(N(C(C(OC3 = CC = C(C = C3F)F) = C2) = O)CC(C)O) = N1)CO
Tokenization	C C (C O) N C 1 = N C = C 2 C (N (C C (O) C) C (C (O C 3 = C C = C (F) C = C 3 F) = C 2) = O) = N 1	C C (N C 1 = N C (O) = C 2 C (N (C (C (O C 3 = C C = C (C = C 3 F) F) = C 2) = O) C C (C) O) = N 1) C O

**Table 4 metabolites-13-00449-t004:** Transformer–baseline and transformer–transfer learning models’ performance on test datasets.

Model	SMILES Validity	Top 1 Accuracy	Top 2 Accuracy	Top 3 Accuracy
Transformer–baseline model	0.69	0.08	0.08	0.10
Transformer–transfer learning model	0.93	0.57	0.63	0.67

**Table 5 metabolites-13-00449-t005:** The performance of our model (Meta-hAOX) on the external data set collected by Zhao et al., compared with the DT_AOX_ model, DTN_AOX_, and three descriptors proposed by Montefiori et al.

Model	SE	SP	ACC	F1
Meta-hAOX	0.77	0.93	0.91	0.77
DT_AOX_	0.50	0.87	0.81	0.47
DTN_AOX_	0.79	0.91	0.89	0.71
NMR shielding	0.71	0.77	0.76	0.50
ESP charge	0.79	0.83	0.82	0.59
Chemical shift	0.71	0.77	0.76	0.50

## Data Availability

All data and code for this article are available at the GitHub repository: https://github.com/mengtinghuang/Meta-hAOX, accessed on 15 December 2022.

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
