# Peer review of "In Silico Prediction of Metabolic Reaction Catalyzed by Human Aldehyde Oxidase"

_metabolites, 2023, doi:10.3390/metabo13030449_

Round 1

Reviewer 1 Report

The study developed a ligand-based model to accelerate the drug design and discovery process to predict human aldehyde oxidase-catalyzed metabolism deposited on a web server, MetahAOX. Compared with molecular docking and quantum chemistry methods, the approach that applied machine learning, graph convolutional network, and sequence-based method, provided a convenient and efficient tool to predict the sites of metabolism of human aldehyde oxidases. It is recommended for publication in the journal “Metabolites.” Below are the suggestions for the improvement of the manuscript.

1.     The keyword “QSAR” was not mentioned in the text. It is suggested to add the full name of QSAR and briefly explain it in the introduction.

2.     Many abbreviations were used in Figure 1. The full name of these abbreviations should be shown in the figure legend. Besides, GNN may need to be corrected because GCN is commonly used to represent graph convolutional network.

3.     It is suggested to add the serial number for the substrates listed in Table S1. The reference could be shown in an additional column at last.  

4.     Section 2.2.1 and Table S2. The authors described that the test set included 168 potential SOMs, 56 of which were true SOM. It meant that non-SOM was 112 in total. However, it showed 53 and 111 for SOM and non-SOM in Table S2; the total was 164. Which information is correct?

5.     The citation, Zheng et al., was cited in the text several times. However, it was not listed in the reference.

6.     The metabolite, 2-mercaptopurine, shown in Table S3, needed to be described in section 4.2.

Reviewer 2 Report

Authors have selected an important enzyme Aldehyde oxidase (AOX) for finding its metabolic reactions in the manuscript entitled “In silico prediction of metabolic reaction catalyzed by human aldehyde oxidase”.

In the manuscript, they have compared traditional machine learning methods, graph convolutional neural network methods, and sequence-based methods with limited data, and proposed a ligand-based model for the metabolism prediction catalyzed by human AOX.

The online server (http://lmmd.ecust.edu.cn/MetahAOX) can predict the sites of metabolism (SOMs) for human AOX. Data size used in this study is relatively small and the given information on server web-site is not sufficient. Authors are suggested to modify these things.

Reviewer 3 Report

In this manuscript, Huang et al. performed in silico analysis of metabolic reactions catalyzed by human aldehyde oxidase (AOX). This study proposed a new model for ligand-based metabolism prediction and compared it with several existing models. The new model showed good prediction rates with high ACC values. I have a few comments on this manuscript:

1, Please provide some explanations on the sample size of the training set is large enough to learn the information. What’s the sample size for similar studies, and how did they evaluate whether the sample size was enough?

2, In Figure 2, the PCA analysis does not provide a clear separation of many molecules on the left, with many of them overlapping. I suggest the authors try tSNE and UMAP and see how that would improve the representation on a 2-dimensional space.

3, What are the main driving factors of the different performances in Figure 3? Is this known that some models perform better than others?

4, In the discussion section, I look forward to more perspectives on how this model would benefit the analysis of metabolic reactions and, in the future, help drug design.

Overall, I believe this is a good paper demonstrating a new model with comprehensive benchmarking and comparison to existing models. Lastly, the web server cannot be opened on my side.
